# Syndromic surveillance during 2022 Uganda Martyrs' commemoration

**Mackline Ninsiima**[1]*, **Mercy W. Wanyana**[1], **Thomas Kiggundu**[1], **Patrick King**[1], **Bernard Lubwama**[2], **Richard Migisha**[1], **Lilian Bulage**[1], **Daniel Kadobera**[1], **Alex Riolexus Ario**[1]

1 Uganda Public Health Fellowship Program, Uganda National Institute of Public Health, Kampala, Uganda,
2 Division of Integrated Epidemiology and Surveillance, Ministry of Health, Kampala, Uganda

* nmackline@musph.ac.ug

## Abstract

Mass gatherings frequently include close, prolonged interactions between people, which presents opportunities for infectious disease transmission. Over 20,000 pilgrims gathered at Namugongo Catholic and Protestant shrines to commemorate 2022 Uganda Martyr's Day. We described syndromes suggestive of key priority diseases particularly COVID–19 and viral hemorrhagic fever (VHF) among visiting pilgrims during May 25–June 5, 2022. We conducted a survey among pilgrims at the catholic and protestant shrines based on signs and symptoms for key priority diseases: COVID–19 and VHF. A suspected COVID–19 case was defined as acute respiratory illness (temperature greater 37.5˚C and at least one sign/symptom of respiratory infection such as cough or shortness of breath) whereas a suspected VHF case was defined as fever >37.5˚C and unexplained bleeding among pilgrims who visited Namugongo Catholic and Protestant shrines from May 25 to June 5, 2022. Pilgrims were sampled systematically at entrances and demarcated zonal areas to participate in the survey. Additionally, we extracted secondary data on pilgrims who sought emergency medical services from Health Management Information System registers. Descriptive analysis was conducted to identify syndromes suggestive of key priority diseases. Among 1,350 pilgrims interviewed, 767 (57%) were female. The mean age was 37.9 (±17.9) years. Nearly all pilgrims 1,331 (98.6%) were Ugandans. A total of 236 (18%) reported ≥1 case definition symptom and 42 (3%) reported ≥2 symptoms. Thirty-nine (2.9%) were suspected COVID–19 cases and three (0.2%) were suspected VHF cases from different regions of Uganda. Among 5,582 pilgrims who sought medical care from tents, 628 (11.3%) had suspected COVID–19 and one had suspected VHF. Almost one in fifty pilgrims at the 2022 Uganda Martyrs' commemoration had at least one symptom of COVID–19 or VHF. Intensified syndromic surveillance and planned laboratory testing capacity at mass gatherings is important for early detection of public health emergencies that could stem from such events.

## Background

According to the World Health Organization (WHO), a mass gathering is an event, either organized or spontaneous, characterized by concentration of people at a specific location for a

**Funding:** The authors received no specific funding for this work.

**Competing interests:** The authors have declared that no competing interests exist.

specific purpose over a set period of time and has the potential to strain planning and response resources of the host country or community [1]. During mass gatherings, overcrowding of attendees and influx of non-local travelers may present favourable and conducive environments for close, prolonged and frequent interactions increasing the opportunities for infectious disease transmission. What is quite challenging is that any adverse health outcome associated with mass gatherings would most likely be magnified by media and political attention. Furthermore, occurrence of a disease outbreak at or during an international mass gathering has an increased potential for spreading to neighboring countries or even globally; thus, the need for reporting under the 2005 International Health Regulations [2]. Influx of people during mass gatherings impacts a strain on existing surveillance and response systems. This presents a challenge to the hosting community or country to strengthen surveillance and response systems during preparation, operational and post-event phases of mass gatherings.

Syndromic surveillance—the utilization of health-related data based on clinical observations and symptoms rather than confirmed diagnosis, can serve as an effective strategy for appropriate real time monitoring, early detection and timely response to public health events during mass gatherings [1, 3, 4]. A provisional diagnosis or a "syndrome" can be established through synthesis of a group of symptoms and clinical observations which consistently occur together. During mass gatherings, syndromic surveillance has been implemented through surveys–recording symptoms, review of medical registers completed by medical teams and utilization of automated alert systems; followed by real-time analysis of data to generate incident reports necessary for informing timely response actions [5]. To date, syndromic surveillance has been utilized in several mass gathering settings: 2002 Winter Olympic Games in Salt Lake City; 2012 Summer Olympic and Paralympic Games in London; 8th Micronesian Games in 2014, 2015 Los Angeles Special Olympic World Games; religious mass gatherings in Southern India; 2016 Grand Magal of Touba in Senegal; and 2016 Arbaeenia mass gathering in Iraq [3, 5–8]. Participatory surveillance enhanced through utilization of digital technologies during mass gatherings has been demonstrated during the 2014 FIFA World Cup [9] and 2016 Olympic Games [10]. Following the COVID-19 pandemic, syndromic surveillance for respiratory tract infections was conducted among pilgrims who attended Hajj in 2021 [11].

Every year, in June, pilgrims from Uganda and neighboring countries gather at Namugongo Catholic and Protestant shrines to commemorate the lives of Uganda Martyrs. In 2020 and 2021, Uganda Martyrs' Day was not physically commemorated due to stringent strategies deployed by Ministry of Health to curb the spread of the COVID–19 pandemic during mass gatherings. In February 2022, the Ugandan Government relaxed the restrictions that had been put in place to control COVID–19 thus approving full economy operation. Following the relaxation of the COVID–19 restrictions, catholic and protestant religious bodies were permitted to organize the commemoration of Uganda Martyr's Day, a historical religious event that calls for a mass gathering at Namugongo Catholic and Protestant shrines from May 25 to June 5, 2022. Due to the distances people travel to attend this event, an infectious disease outbreak that starts during this mass gathering has high potential to spread to neighboring districts or even to other countries.

During the commemoration of the Uganda Martyrs in 2022, the Ministry of Health in collaboration with the Uganda Catholic and Protestant Medical Bureaus provided health services, including on-site emergency medical services in designated tents from May 25 to June 5, 2022. The Ministry of Health also provided Health Management Information System registers where data for pilgrims who sought medical care were captured by the medical teams to achieve harmonized reporting from the different institutions. Additionally, the National Rapid Response Team of the Ministry of Health were assigned to conduct syndromic surveillance for

key priority diseases during the event. Therefore, we described syndromes suggestive of key priority diseases among visiting pilgrims from May 25 to June 5, 2022.

## Methods

### Study setting and study population

This assessment was conducted among over 20,000 visiting pilgrims from Uganda and neighboring countries gathered at Namugongo Catholic and Protestant shrines located in Namugongo Division, Wakiso District, Uganda. On–site emergency medical services were provided in designated tents from May 25 to June 5, 2022 by medical teams comprising emergency medicine specialists, doctors, nurses, laboratory attendants, and ambulance teams from Ministry of Health, Mulago National Referral Hospital, St. Francis Hospital Nsambya, Uganda Martyrs Hospital Rubaga, Uganda People's Defence Forces, Uganda Police Force, Uganda Red Cross Society, St. John's Ambulance, Holy Family Virika Hospital, Mengo Hospital, Zia Angellina Health Centre, and St. Stephens Hospital.

### Data collection

We utilized two different methods for data collection. First, we conducted a survey among pilgrims at the Catholic and Protestant shrines based on signs and symptoms for key priority diseases from June 2–5, 2022. The data collection tool was developed in KoboCollect based on signs and symptoms for selected priority diseases: COVID–19 and VHFs inclusive of Ebola Virus Disease, Crimean Congo Hemorrhagic Fever, Yellow Fever, Rift Valley Fever, and Marburg Hemorrhagic Fever. Signs and symptoms investigated were based on suspect case definitions as per the National Technical Guidelines for Integrated Disease Surveillance and Response. Signs and symptoms under investigation were: fever (temperature $>37.5°C$), cough, headache, generalized body weakness, shortness of breath, jaundice, fainting or sudden collapse, and unexplained bleeding. Any other signs and symptoms reported by the participants were also recorded by the surveillance officers.

We sampled systematically every 10th pilgrim in the line at main entrance gates. Other pilgrims were selected randomly from demarcated zonal areas. Verbal informed consent was obtained from participants prior to interviews. Overall, surveillance officers from Makerere University School of Public Health interviewed 1,350 pilgrims who voluntarily participated in the survey. Survey data were downloaded in the Excel (.xls) format from the KoboCollect server and processed for analysis. Second, we conducted records review based on the on-site emergency medical services provided at the Catholic and Protestant shrines from May 25 to June 5, 2022. We extracted all the available data on 5,582 pilgrims who sought medical care from Health Management Information System registers for review including age, sex, district of residence, signs and symptoms or provisional diagnosis.

### Data analysis

We conducted univariate data analysis using Epi Info 7 software (CDC, Atlanta, USA) to obtain frequencies of demographic characteristics and syndromes suggestive of public health emergencies among pilgrims who participated in the survey or sought care from the medical tents. Age categories were based on the ranking utilized by Uganda Bureau of Statistics (UBOS). Only syndromes suggestive of key priority diseases were of interest to the investigative team. At analysis phase, a suspected COVID–19 case was defined as acute respiratory illness (temperature greater $37.5°C$ and at least one sign/symptom of respiratory infection such as cough or shortness of breath) whereas a suspected VHF case was defined as fever $>37.5°C$

and unexplained bleeding among pilgrims who visited Namugongo Catholic and Protestant shrines from May 25 to June 5, 2022.

### Ethical considerations

This investigation was in response to an annually commemorated mass gathering and was therefore determined to be non-research; accordingly (in reference to the memorandum of understanding, the activity was waived the process of full Institutional Review Board). In our country, the Uganda National Institute of Public Health (UNIPH), falls under the Ministry of Health (MoH); in essence, the UNIPH is a subordinate authority to MoH. The MoH gave authority and directive to conduct syndromic surveillance during this religious event, then UNIPH approved the study and all the study protocols that were used. All methods were performed in accordance with the approval and administrative clearance without any ethical breach.

Verbal consent in English and the local language was sought from pilgrims before participation in the survey. Systematically sampled participants were informed about the rationale of the survey and that their participation was voluntary without any negative consequences in case they refused. In accordance with the study protocol approved by UNIPH, there was a question in the screening section of the survey questionnaire inquiring as to whether the systematically sampled participant verbally consented; and for those who did not, they were skipped to the next 10[th] pilgrim in the line at main entrance gates. Pilgrims were assigned unique identifiers instead of using their names to protect the confidentiality of the respondents. Administrative clearance to extract patient data from Health Management Information System registers was obtained from the Ministry of Health. All methods were performed in accordance with the approval and administrative clearance.

## Results

### Characteristics of pilgrims who participated in the survey during the Uganda Martyrs' commemoration mass gathering, June 2–5, 2022

Among the 1,350 pilgrims who participated in the survey, 767 (56.8%) were females. Nearly all pilgrims 1,331 (98.6%) were Ugandans. The majority of pilgrims 1,157 (85.7%) visited the Catholic shrine (Table 1).

### Characteristics of pilgrims sought medical care from medical tents during the Uganda Martyrs' commemoration mass gathering, May 25–June 5, 2022

Among the 5,582 pilgrims who sought medical care from the medical tents 3,901 (70.1%) were females whereas 1,521 (27.5%) were aged ≥50 years (Table 2). Age was not recorded among 57 pilgrims who sought medical care from the medical tents.

### Suspected priority diseases

Among the 1,350 pilgrims who participated in the survey, 653 (48.4%) reported at least one sign or symptom during their visit to the Catholic and Protestant shrines. Of these, 236 (18%) reported ≥1 suspected COVID–19 and VHF case definition signs and symptoms and 25 (2%) reported ≥2 symptoms (Fig 1). Thirty-nine (2.9%) were suspected COVID–19 cases and three (0.2%) were suspected VHF cases from different regions of Uganda, two were bleeding from the nose and one had bloody vomitus and urine.

**Table 1. Characteristics of pilgrims who participated in the survey during the Uganda Martyrs' commemoration mass gathering, June 2–5, 2022.**

| Characteristic | Frequency (n = 1,350) | Percentage (%) |
|---|---|---|
| Age* | *Median age (IQR)*: 35 (25–49) | *Mean Age (SD)*: 37.9 (17.6) |
| <18 years | 85 | 6.3 |
| 18–29 years | 409 | 30.3 |
| 30–39 years | 284 | 21.0 |
| 40–49 years | 237 | 17.6 |
| ≥50 years | 335 | 24.8 |
| Sex | | |
| Male | 583 | 43.2 |
| Female | 767 | 56.8 |
| Country of residence | | |
| Uganda | 1,331 | 98.6 |
| Kenya | 9 | 0.7 |
| South Sudan | 4 | 0.3 |
| Rwanda | 2 | 0.2 |
| Democratic Republic of Congo | 1 | 0.1 |
| Nigeria | 3 | 0.2 |
| Religious site visited | | |
| Catholic shrine | 1,157 | 85.7 |
| Protestant shrine | 153 | 14.3 |

Among 5,582 pilgrims who sought care at the medical tents, 3,796 records specified the presenting signs and symptoms whereas 1,786 records did not have specified signs and symptoms but only had a provisional diagnosis based on the clinician's assessment. Of 3,796 records, 629 pilgrims reported atleast 2 symptoms suggestive of key priority diseases. Of these, 628 (11.3%) had suspected COVID–19 and one had suspected VHF with bloody vomitus (Fig 2).

## Discussion

In this study, we described syndromes suggestive of key priority diseases among visiting pilgrims from May 25–June 5, 2022 to inform future planning for mass gatherings in Uganda.

**Table 2. Characteristics of pilgrims who sought medical care from medical tents during the Uganda Martyrs' commemoration mass gathering, May 25–June 5, 2022.**

| Characteristics | Frequency (n = 5,582) | Percentage (%) |
|---|---|---|
| Age* (n = 5,525) | *Median age (IQR)*: 38 (25–51) | *Mean Age (SD)*: 38.6 (18.0) |
| <18 years | 726 | 13.1 |
| 18–29 years | 1,143 | 20.7 |
| 30–39 years | 1,051 | 19.0 |
| 40–49 years | 1,084 | 19.6 |
| ≥50 years | 1,521 | 27.5 |
| Sex | | |
| Male | 1,668 | 29.9 |
| Female | 3,914 | 70.1 |
| Chronic illness | | |
| Diabetes | 33 | 0.6 |
| Hypertension | 111 | 2.0 |

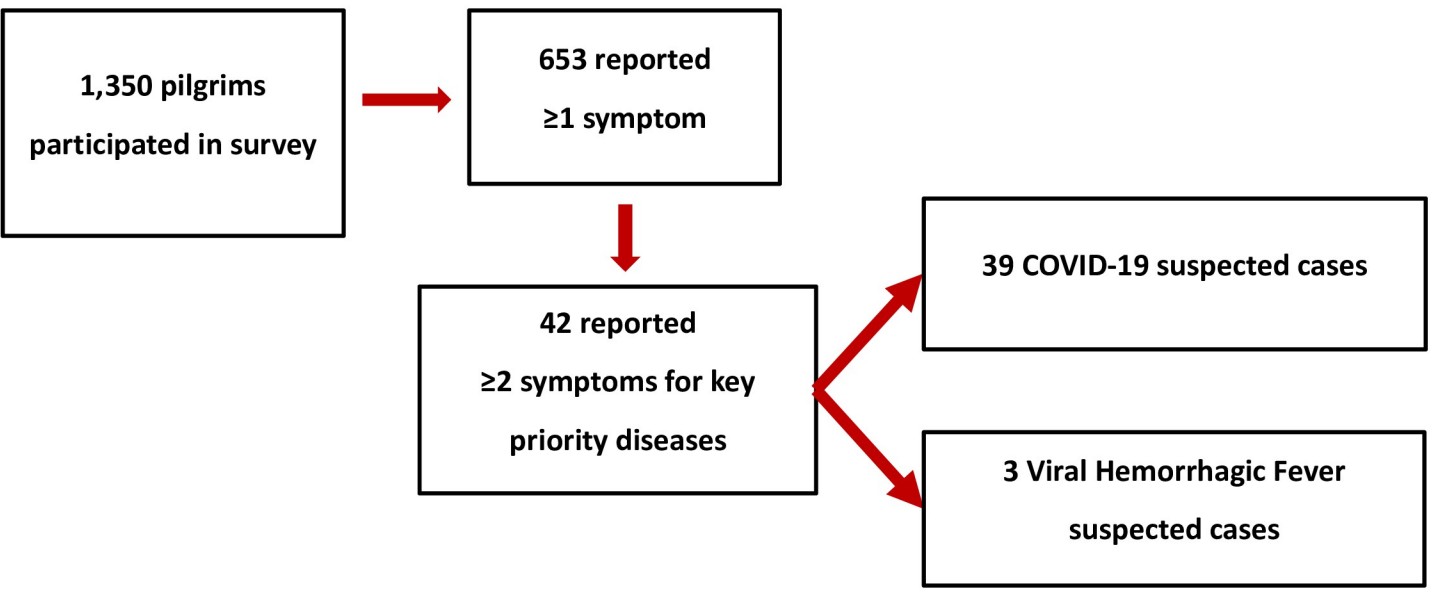

**Fig 1. Schema showing suspected priority diseases among pilgrims who participated in the survey during the Uganda Martyrs' commemoration mass gathering, May 25–June 5, 2022.**

We identified 4 VHF and 667 COVID–19 suspected cases through syndromic surveillance during the 2022 Uganda Martyrs' commemoration. Similarly, previous incidences of outbreaks have been reported following festive, religious and sporting-related mass gatherings including a COVID–19 outbreak after festivities in Spain; outbreaks of diarrheal diseases

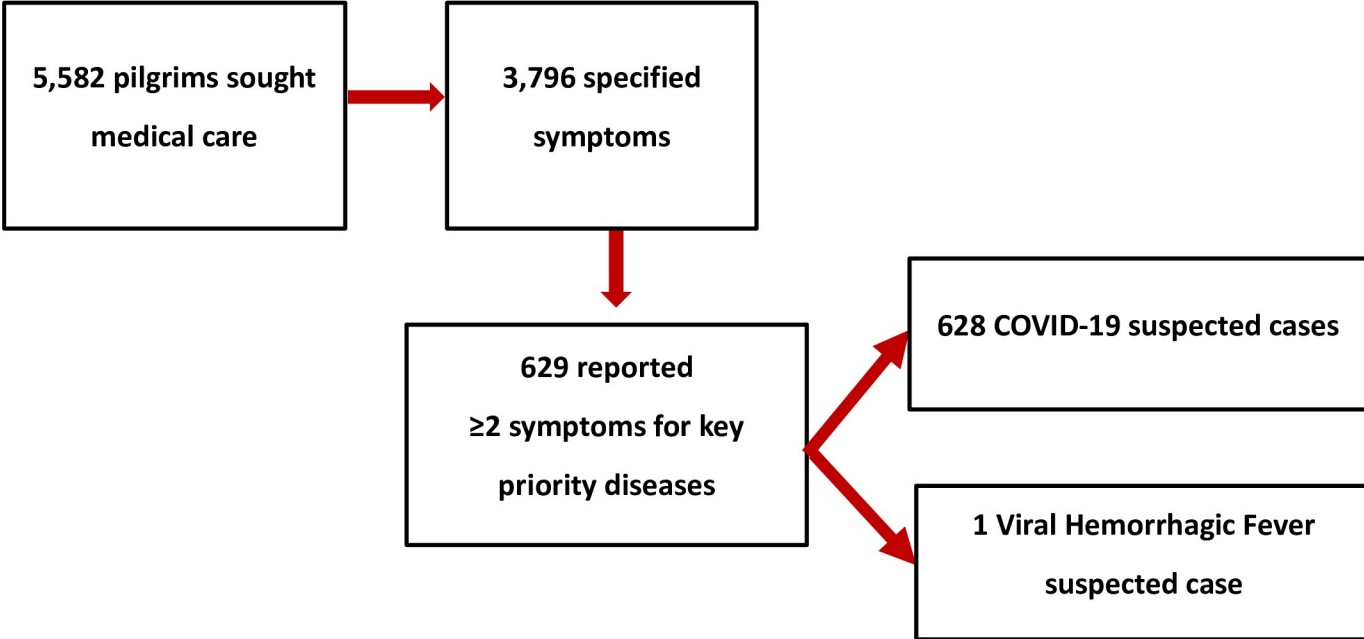

**Fig 2. Schema showing suspected priority diseases among pilgrims who sought medical care from medical tents during the Uganda Martyrs' commemoration mass gathering, May 25–June 5, 2022.**

during the 2019 Hijja pilgrimage in Saudi Arabia; mumps following festive activities in Austria and Spain; measles after an international youth sporting event in United States of America; meningococcal disease associated with the 23rd World Scout Jamboree gathering in Japan; and influenza H1N1 outbreaks after music festivals in Belgium and Hungary [1, 12–17]. Mass gatherings have been highly characterized as transmission sites for infectious diseases due to close proximity and rapid and possibility of dissemination of infectious agents after the mass dispersion to different locations [18].

During the COVID–19 pandemic, mass gatherings were highlighted among the sources of transmission due to overcrowding and close interaction between attendees. In 2020, a social gathering at a church in Omoro District, Northern Uganda provided an opportunity for a COVID–19 superspreading event [19]. A recent systematic review reported that religious gatherings in places of worship were vital in COVID–19 transmission accounting for over 50 worship related clusters especially during the first wave of the pandemic [20]. Mass gatherings have been significantly associated with COVID–19 transmission [12, 21–23]. It should also be noted that risk of transmission of infectious diseases could be partly influenced by the type, venue, location and demographics of participants who attend the mass gatherings [24–26]. Evidence of COVID–19 transmission during mass gatherings has also been reported in Malaysia during the Sri Petaling Moslem Missionary Movement [27]. Due to such scenarios, the WHO published interventions which should be implemented to mitigate the spread of COVID–19 during mass gatherings [28].

Limited evidence has been presented on the incidence of VHFs during mass gatherings. However, there is still need for great vigilance since most VHFs are largely characterized by person to person transmission which could be highly favored by the close contact between attendees during mass gatherings. Experience from mass gathering events held during the West African Ebola epidemic illustrates that these events can be held safely provided interventions are instituted for enhanced surveillance and response systems for infectious diseases [29]. This underscores the urgent need of implementing effective measures to mitigate the spread of any VHF during such annual mass gatherings.

Evidence generated from this assessment indicates potential health risks linked to mass gatherings. Uganda should conduct intra and after-action reviews of mass gatherings to identify strengths and weaknesses of existing surveillance and response systems, identify areas for improvement, and provide actionable recommendations. This could serve as a learning opportunity to enhance the country's readiness and response capabilities for future mass gatherings. Integration of simulation exercises will be essential to evaluate existing protocols, identify gaps and refine strategies for managing large-scale events. Conducting risk assessments prior to mass gatherings could guide the development of tailored strategies for mitigating health risks, including the allocation of resources for syndromic surveillance, healthcare infrastructure, and emergency response.

Based on the findings, policy makers should increase funding, conduct trainings, and construct infrastructure, enhancing the public health system's ability to monitor and respond to potential outbreaks during mass gatherings. Future studies should focus on validating syndromic surveillance methods for mass gatherings, exploring the effectiveness of specific interventions implemented in response to such events, and assessing the long-term health impacts on both attendees and the surrounding communities. Furthermore, there is need to investigate the socio-economic and cultural factors influencing the transmission of infectious diseases during mass gatherings, providing a more holistic understanding of the associated risks.

However, there were only seventeen trained surveillance officers despite the masses at the Namugongo Protestant and Catholic shrines, who started administering the survey questionnaire on June 2, 2022 instead of having commenced on the May 25, 2022, at the time when

pilgrims started gathering. Therefore, it was difficult to generalize the findings on the signs and symptoms for selected priority diseases to the entire population that converged during the 2022 Uganda Martyrs' commemoration. Additionally, 1,786 out of 5,582 records did not have specified signs and symptoms but only had a provisional diagnosis based on the clinician's assessment. We could not categorize these pilgrims under any of the key priority diseases since they did not have specified signs and symptoms; which could have underestimated the syndromes suggestive of key priority diseases. Furthermore, this study was largely a descriptive study which did not test any hypotheses.

## Conclusion

Almost one in fifty pilgrims at the 2022 Uganda Martyrs' commemoration had symptoms of COVID–19 or VHF. Unfortunately, none of the suspected COVID–19 or VHF cases were tested and we do not know what condition they had. While we have no evidence that the suspected VHF cases had any link to the 2022 Ebola Virus Disease outbreak in Uganda, it is clear from these findings that a surveillance system at mass gatherings and the ability to actively respond to possible cases is critical. It is important for us to prioritize intensified syndromic surveillance during mass gatherings to ensure that we reduce the risk for an outbreak at mass gatherings in Uganda and reduce the impact if one should occur. Furthermore, there is utmost need to set up isolation facilities for any suspected cases and provide laboratory testing capacity to facilitate early detection and response to priority key diseases that could stem from such events. Ultimately, evidence generated from the 2022 Uganda Martyrs' Commemoration underscores the paramount importance of proactive public health planning and response strategies for mass gatherings. By diligently incorporating the valuable lessons learnt from this event, Uganda may not only enhance but also expand its capacity to protect the public during similar occasions in the future.

## Supporting information

**S1 File. Medical records.**
(XLS)

**S2 File. Survey data.**
(XLS)

**S1 Checklist. STROBE statement—Checklist of items that should be included in reports of observational studies.**
(DOCX)

## Acknowledgments

The authors would like to thank medical teams from Ministry of Health, Mulago National Referral Hospital, St. Francis Hospital Nsambya, Uganda Martyrs Hospital Rubaga, Uganda People's Defence Forces, Uganda Police Force, Uganda Red Cross Society, St. John's Ambulance, Holy Family Virika Hospital, Mengo Hospital, Zia Angellina Health Centre, and St. Stephens Hospital who provided onsite emergency medical services at the Catholic and Protestant shrines. Additional thanks to surveillance officers from Makerere University School of Public Health (MakSPH) who administered survey questionnaires to pilgrims at the Catholic and Protestant shrines.

## Author Contributions

**Conceptualization:** Mackline Ninsiima, Mercy W. Wanyana, Thomas Kiggundu, Patrick King, Bernard Lubwama, Richard Migisha, Lilian Bulage, Daniel Kadobera, Alex Riolexus Ario.

**Data curation:** Mackline Ninsiima.

**Formal analysis:** Mackline Ninsiima, Mercy W. Wanyana, Thomas Kiggundu, Patrick King, Bernard Lubwama, Richard Migisha, Lilian Bulage, Daniel Kadobera, Alex Riolexus Ario.

**Investigation:** Mackline Ninsiima, Mercy W. Wanyana, Thomas Kiggundu, Patrick King, Bernard Lubwama, Richard Migisha, Lilian Bulage, Daniel Kadobera, Alex Riolexus Ario.

**Methodology:** Mackline Ninsiima, Mercy W. Wanyana, Thomas Kiggundu, Patrick King, Bernard Lubwama, Richard Migisha, Lilian Bulage, Daniel Kadobera, Alex Riolexus Ario.

**Project administration:** Mackline Ninsiima.

**Supervision:** Mackline Ninsiima, Mercy W. Wanyana, Thomas Kiggundu, Patrick King, Bernard Lubwama, Richard Migisha, Lilian Bulage, Daniel Kadobera, Alex Riolexus Ario.

**Validation:** Mackline Ninsiima, Mercy W. Wanyana, Thomas Kiggundu, Patrick King, Bernard Lubwama, Richard Migisha, Lilian Bulage, Daniel Kadobera, Alex Riolexus Ario.

**Visualization:** Mackline Ninsiima, Mercy W. Wanyana, Thomas Kiggundu, Patrick King, Bernard Lubwama, Richard Migisha, Lilian Bulage, Daniel Kadobera, Alex Riolexus Ario.

**Writing – original draft:** Mackline Ninsiima, Mercy W. Wanyana, Thomas Kiggundu, Patrick King, Bernard Lubwama, Richard Migisha, Lilian Bulage, Daniel Kadobera, Alex Riolexus Ario.

**Writing – review & editing:** Mackline Ninsiima, Mercy W. Wanyana, Thomas Kiggundu, Patrick King, Bernard Lubwama, Richard Migisha, Lilian Bulage, Daniel Kadobera, Alex Riolexus Ario.

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
