## [Decision Letter · Decision Letter 0]

3 Jul 2023

PGPH-D-23-00954

Syndromic surveillance during 2022 Uganda martyrs’ commemoration

Dear Dr. Ninsiima,

Thank you for submitting your manuscript to PLOS Global Public Health. After careful consideration, we feel that it has merit but does not fully meet PLOS Global Public Health’s publication criteria as it currently stands. Therefore, we invite you to submit a revised version of the manuscript that addresses the points raised during the review process.

Please note that we have only been able to secure a single reviewer to assess your manuscript. We are issuing a decision on your manuscript at this point to prevent further delays in the evaluation of your manuscript. Please be aware that the editor who handles your revised manuscript might find it necessary to invite additional reviewers to assess this work once the revised manuscript is submitted. However, we will aim to proceed on the basis of this single review if possible.

The reviewer has identified several ways in which the scientific rigor and reproducibility of the research, as well as its overall contribution to the field, could be enhanced.

The reviewer has also commented on the statement provided in your manuscript regarding ethical approval. We agree with the reviewer that it is clear that your study did involve human participants, and is therefore - in contrast to your statement - human subject research. However, we also recognize that anonymized public health surveillance can be considered exempt from ethical approval by an institutional review board or equivalent research ethics committee. Please revise your statement in the 'Ethical considerations' section of your manuscript text to clarify this point. You may wish to provide a letter from your institutional review board or equivalent ethics committee confirming this study was exempt from ethical approval; please upload a copy of any such letter as an 'Other' file with your revised manuscript files.

We look forward to receiving your revised manuscript.

Kind regards,

Hugh Cowley

Staff Editor

Journal Requirements:

1. In the ethics statement in the Methods, you have specified that verbal consent was obtained. Please provide additional details regarding how this consent was documented and witnessed, and state whether this was approved by the IRB

2. Please ensure that Funding Information and Financial Disclosure Statement are matched.

3. In the Funding Information you indicated that no funding was received. Please revise the Funding Information field to reflect funding received.

4. Please provide separate figure files in .tif or .eps format only and remove any figures embedded in your manuscript file. Please also ensure all files are under our size limit of 10MB.

5. We have noticed that you have uploaded Supporting Information files, but you have not included a list of legends. Please add a full list of legends for your Supporting Information files after the references list. 

6. In the online submission form, you indicated that "The datasets upon which our findings are based belong to the Uganda Public Health Fellowship Program. For confidentiality reasons the datasets are not publicly available. However, the data sets can be availed upon reasonable request from the corresponding author and with permission from the Uganda Public Health Fellowship Program". All PLOS journals now require all data underlying the findings described in their manuscript to be freely available to other researchers, either 1. In a public repository, 2. Within the manuscript itself, or 3. Uploaded as supplementary information.

Additional Editor Comments (if provided):

Reviewers' comments:

Reviewer's Responses to Questions

**Comments to the Author**

1. Does this manuscript meet PLOS Global Public Health’s publication criteria? Is the manuscript technically sound, and do the data support the conclusions? The manuscript must describe methodologically and ethically rigorous research with conclusions that are appropriately drawn based on the data presented.

Reviewer #1: Yes

2. Has the statistical analysis been performed appropriately and rigorously?

Reviewer #1: No

3. Have the authors made all data underlying the findings in their manuscript fully available (please refer to the Data Availability Statement at the start of the manuscript PDF file)?

Reviewer #1: No

4. Is the manuscript presented in an intelligible fashion and written in standard English?

Reviewer #1: Yes

5. Review Comments to the Author

Reviewer #1: General comment:

The manuscript presents a compelling analysis of syndromic surveillance at a large gathering event, which is an important contribution to the literature on mass gatherings and disease outbreaks. However, several areas of the manuscript require additional detail and clarification to enhance the scientific rigor and reproducibility of the research, as well as to improve the paper's contribution to the scholarly discourse.

Specific comments:

Introduction:

The authors should ensure that they fully acknowledge the contributions of seminal papers on syndromic surveillance during mass gatherings. References such as https://doi.org/10.2196/publichealth.7313 and https://doi.org/10.2196/16119 should be discussed and included in the paper. The authors should also consider including discussions on notable syndromic surveillance efforts such as the ones during Hajj. One suggested paper on this subject is https://bmcinfectdis.biomedcentral.com/articles/10.1186/s12879-022-07559-0.

The study's objective should also be clearly stated in the introduction. Readers should not have to infer the aim of the research.

Methods:

The symptoms selected for surveillance could be reconsidered. The use of 'flu' as a symptom could be misleading, as syndromic surveillance typically employs broader categories such as Acute Respiratory Infections (ARIs).

The paper's handling of ethical considerations needs clarification. It is contradictory to state that the study did not involve human subjects while also describing the collection of verbal informed consent from participants.

Results and Discussion:

The data analysis also refers to 'flu' as a symptom. Consistent with my earlier comment, I would recommend removing this term and instead categorizing under ARIs. It would be prudent to consider that the symptoms screened for might be suggestive of several respiratory viruses, not just COVID-19.

The age brackets used in tables 1 and 2 should be clearly explained in the methodology section. The parameters for grouping age can significantly impact the study's findings, and thus need to be transparent.

The paper could benefit from more advanced statistical analysis. The study, in its current form, is largely descriptive and fails to test any hypotheses. The limitations section should acknowledge this, and any other shortcomings of the study that were not previously mentioned.

Overall, while the topic and approach are relevant and interesting, the manuscript requires significant clarification and augmentation to meet publication standards. Addressing these comments will make a stronger contribution to the discourse on syndromic surveillance during mass gatherings and potentially influence best practices during such events.

6. PLOS authors have the option to publish the peer review history of their article (what does this mean?). If published, this will include your full peer review and any attached files.

**Do you want your identity to be public for this peer review?** For information about this choice, including consent withdrawal, please see our Privacy Policy.

Reviewer #1: **Yes: **Onicio Batista Leal Neto

---

## [Editor Report · Decision Letter 1]

25 Sep 2023

PGPH-D-23-00954R1

Syndromic surveillance during 2022 Uganda martyrs’ commemoration

Dear Dr. Ninsiima,

Thank you for submitting your manuscript to PLOS Global Public Health. After careful consideration, we feel that it has merit but does not fully meet PLOS Global Public Health’s publication criteria as it currently stands. Therefore, we invite you to submit a revised version of the manuscript that addresses the points raised during the review process.

Please review the editor's comments below and submit your revised manuscript by Oct 25 2023 11:59PM. If you will need more time than this to complete your revisions, please reply to this message or contact the journal office at globalpubhealth@plos.org. Please include the following items when submitting your revised manuscript:

We look forward to receiving your revised manuscript.

Kind regards,

Sanjana Ravi, PhD, MPH

Academic Editor

Journal Requirements:

Additional Editor Comments (if provided):

Thank you for addressing the comments raised by the previous reviewer, which has improved the manuscript. However, a few more revisions are needed before this manuscript can be considered for publication.

1) The previous reviewer requested that you not list "flu" as a symptom of COVID-19, which is a concern that I also share, as influenza is not a symptom of COVID-19 -- did you mean to say "influenza-like illness?" "Acute respiratory illness" would also make sense instead of "flu," per Reviewer 1's suggestion.

Furthermore, per the case definition provided in Uganda's IDSR guidelines (see Pg 418), there is no mention of "flu" as a criterion for COVID-19: https://www.afro.who.int/sites/default/files/2021-09/2_Uganda%203rd%20IDSR%20Tech%20Guideline_PrintVersion_10Sep2021.pdf

As such, please review the analysis to ensure that the data presented are in line with the correct case definition. Please also cite the IDSR guidelines so readers can look up case definitions, or if a different case definition of COVID-19 was used, please provide a citation in the text.

2) The discussion and conclusion would benefit from some consideration of the implications of this analysis for public health policymaking and practice in Uganda. The conclusion acknowledges the importance of prioritizing intensified syndromic surveillance, but some elaboration would be beneficial. Consider adding a few sentences or a paragraph about how Uganda might improve public health planning for future mass gathering events (e.g. intra/after-action review of the 2022 Martyrs' Commemoration, simulation exercises, risk assessments) and/or engage with policymakers to ensure that more resources are allocated toward syndromic surveillance. Your thoughts on future studies/analyses on this topic would also be beneficial.

3) In Line 177 of the revised manuscript, there is a reference to "Church X." I understand that the church may need to be kept anonymous, but "Church X" reads a bit awkwardly. I suggest stating "In 2020, a social gathering at a church in Omoro District, Northern Uganda provided an opportunity for a COVID-19..." instead.
---

## [Editor Report · Decision Letter 2]

1 Dec 2023

Syndromic surveillance during 2022 Uganda martyrs’ commemoration

PGPH-D-23-00954R2

Dear Ms. Ninsiima,

We are pleased to inform you that your manuscript 'Syndromic surveillance during 2022 Uganda martyrs’ commemoration' has been provisionally accepted for publication in PLOS Global Public Health.

Best regards,

Sanjana J. Ravi, PhD, MPH

Academic Editor
